# Association Between FOXP3 and OX40 Expression in Adult T-Cell Leukemia Cells

**DOI:** 10.3390/v17111445

**Published:** 2025-10-30

**Authors:** Mariko Mizuguchi, Yoshiaki Takahashi, Reiko Tanaka, Naoki Imaizumi, Akio Yamashita, Nobuko Matsushita, Takuya Fukushima, Yuetsu Tanaka

**Affiliations:** 1Laboratory of Immunology, Department of Medical Technology, School of Life and Environmental Science, Azabu University, Sagamihara 252-5201, Japan; matsushita@azabu-u.ac.jp; 2Department of Investigative Medicine, Graduate School of Medicine, University of the Ryukyus, Ginowan 901-2720, Japan; ytakah3@cs.u-ryukyu.ac.jp (Y.T.); yamasita@phar.kindai.ac.jp (A.Y.); 3Laboratory of Hematoimmunology, School of Health Sciences, Faculty of Medicine, University of the Ryukyus, Ginowan 901-2720, Japan; reiko_tanaka@s5.dion.ne.jp (R.T.); fukutaku@cs.u-ryukyu.ac.jp (T.F.); 4Laboratory of Clinical Physiology, School of Health Sciences, Faculty of Medicine, University of the Ryukyus, Ginowan 901-2720, Japan; imaizumi@cs.u-ryukyu.ac.jp; 5Laboratory of Molecular Cell Biology, Department of Pharmaceutical Sciences, Kindai University, Higashi-Osaka 577-8502, Japan

**Keywords:** HTLV-1, ATL, FOXP3, OX40, OX40L

## Abstract

Since forkhead box P3 (FOXP3) is a hallmark of regulatory T (Treg) cells, the expansion of FOXP3^+^ adult T-cell leukemia/lymphoma (ATL) cells is believed to contribute to immune suppression and the pathogenesis of ATL. However, the mechanisms underlying the expansion of FOXP3^+^ ATL cells remain unclear. OX40, a co-stimulatory molecule, is expressed in ATL cells, and OX40 signaling has been shown to promote the differentiation and proliferation of Treg cells in mouse models. To investigate the mechanisms driving the expansion of FOXP3^+^ ATL cells, we examined the expression of OX40 and its ligand, OX40L. Our findings revealed that OX40 expression was elevated in patients with ATL and with a high frequency of FOXP3^+^ ATL cells. Flow cytometric analysis of peripheral blood mononuclear cells (PBMCs) from patients with acute ATL cultured for 18 h demonstrated that FOXP3^−^ and FOXP3^+^ cells predominantly expressed OX40L and OX40, respectively. Furthermore, small interfering RNA-mediated FOXP3 knockdown in HTLV-1-infected cell lines increased OX40L expression. These results suggest that interactions between FOXP3^−^ OX40L^+^ cells and FOXP3^+^ OX40^+^ cells may promote the proliferation of FOXP3^+^ ATL cells.

## 1. Introduction

Adult T-cell leukemia/lymphoma (ATL) is an aggressive malignant disease of mature T cells caused by human T-cell leukemia virus type-1 (HTLV-1) infection [1,2]. The number of HTLV-1 carriers is estimated to be 5–10 million worldwide [3]. Although most individuals infected with HTLV-1 remain asymptomatic carriers throughout their lifetime, approximately 5% develop ATL [4]. The accumulation of various genetic mutations and epigenetic changes in HTLV-1-infected cells can lead to leukemogenesis [5,6]. However, the detailed mechanisms underlying the development of ATL remain unclear.

The transcription factor forkhead box P3 (FOXP3) is a marker of regulatory T (Treg) cells that maintain self-tolerance and immune homeostasis. Treg cells suppress activated immune responses through the expression of immune checkpoint molecules such as cytotoxic T lymphocyte antigen 4 (CTLA-4) and immunosuppressive cytokines such as interleukin (IL)-10 and transforming growth factor (TGF)-β. The subsets of Treg cells include naturally occurring thymus-derived Treg (tTreg) cells and peripherally derived Treg (pTreg) cells, which are induced from naïve T cells in peripheral lymphoid tissues, particularly when T cells recognize intestinal antigens in gut-associated lymphoid tissues [7,8]. In vitro, induced Treg (iTreg) cells can be induced by stimulating naïve CD4^+^ T cells with an anti-CD3 antibody in the presence of IL-2 and TGF-β [9,10]. Approximately 60% of ATL cases have been reported to show upregulation of FOXP3 in ATL cells from peripheral blood (PB) [11]. Transgenic mice expressing the antisense viral gene HTLV-1 bZIP factor (HBZ) in CD4^+^ T cells develop lymphomas expressing Foxp3 [12]. Since FOXP3^+^ ATL cells possess an immunosuppressive function [13], aberrant expansion of FOXP3^+^ cells has been thought to be involved in the development of ATL. However, the mechanisms underlying FOXP3^+^ HTLV-1-infected cell expansion in patients with ATL are unknown.

ATL cells express various cell surface molecules, such as OX40, CD25, CD30, and tumor necrosis factor receptor type-2 (TNFR2), which are involved in cell growth and survival [14,15,16,17,18]. OX40 is a member of the TNF receptor superfamily and functions as a costimulatory molecule and is typically expressed on activated human effector T cells following T-cell receptor (TCR) stimulation through antigen presentation. It signals primarily through the nuclear factor (NF)-κB pathway, enhancing T-cell activation and promoting effector functions [19]. Engagement of OX40 in the presence of anti-CD3 stimulation has been shown to notably enhance the proliferation of CD4^+^ T cells [20]. Because the expression of soluble OX40 is elevated in the plasma of patients with acute-type ATL, we have reported its potential as a biomarker [21]. One mechanism underlying the induction of the *OX40* gene is mediated by the oncogenic viral protein Tax [22]. We had also previously identified the OX40 ligand, OX40L (gp34), as a molecule expressed on HTLV-1-infected cells [23,24]. The expression of OX40L is also induced by Tax through the NF-κB pathway [25].

Unlike in humans, mouse Treg cells constitutively express OX40. Stimulation of CD4^+^ CD25^+^ T cells with soluble OX40L has been shown to selectively induce the proliferation of Foxp3^+^ Treg cells [26]. Thus, in individuals infected with HTLV-1, enhanced expression of the OX40L/OX40 axis is expected to selectively promote the proliferation of FOXP3^+^ HTLV-1-infected cells through their activation.

To elucidate the mechanisms underlying the expansion of FOXP3^+^ HTLV-1-infected cells, the expression of OX40 and OX40L in FOXP3^+^ cells from patients with ATL was examined. Our results revealed that FOXP3^+^ cells expressed OX40, while FOXP3^−^ cells expressed OX40L. The expression of OX40L was regulated by FOXP3. These findings suggested that FOXP3 contributes to the proliferation of FOXP3^+^ ATL cells by regulating the OX40/OX40L system.

## 2. Materials and Methods

### 2.1. Materials

Phycoerythrin (PE) or fluorescein isothiocyanate (FITC)-conjugated FOXP3 (clone 206D) and PE/Cyanin 7-conjugated anti-human CD4 (RPA-T4) antibodies were purchased from BioLegend (San Diego, CA, USA). Mouse anti-OX40 (B-7B5), anti-OX40L (HD1), anti-Tax (Lt-4), and anti-CD25 (clone H-40) monoclonal antibodies (mAbs) were developed and purified in our laboratory [27,28,29,30]. These mAbs were labeled with FITC or HyLite Fluor 647 using commercial labeling kits (DOJINDO Molecular Technologies, Kumamoto, Japan).

### 2.2. Cells and Cell Culture

Samples from HTLV-1-infected individuals living in Okinawa Prefecture and healthy donors were obtained with the approval of the Internal Review Committee of the University of the Ryukyus and Azabu University (permit numbers: 23-2205-01-00-00, 11-319-07-04-00, and 157). The participants were enrolled after obtaining written informed consent. All the experiments were performed in accordance with the principles of the Declaration of Helsinki. The disease status of HTLV-1-infected individuals was classified according to Shimoyama’s criteria [31]. Peripheral blood mononuclear cells (PBMCs) were isolated by density-gradient centrifugation using a lymphocyte separation solution (Nacalai Tesque, Kyoto, Japan). The IL-2-dependent HTLV-1-infected cell line YT#1 was established from natural Treg cells isolated from a healthy donor by co-cultivation with the HTLV-1-producing B cell line ATL-040. ATL-026i was derived from an acute-type patient, as previously described [32]. YT#1 and ATL-026i were cultured in RPMI-1640 medium containing 10% fetal calf serum (FCS) and 20 U/mL of IL-2 (Shionogi, Osaka, Japan). Induction of HTLV-1 antigens in the ATL-072, ATL-145, and ATL-224 samples was achieved by culturing the cells in a medium containing IL-2 at a cell concentration of 2 × 10^6^ cells/mL for 18 h.

### 2.3. Flow Cytometric Analysis

Cells were fixed with 1% paraformaldehyde for 10 min at room temperature and washed with FACS buffer (phosphate-buffered saline [PBS] containing 0.2% bovine serum albumin [BSA] and 0.1% sodium azide). The cells were resuspended in 0.2% digitonin in FACS buffer, stained with labeled anti-FOXP3, anti-OX40, anti-OX40L, anti-Tax, anti-CD25, and anti-CD4 antibodies for 30 min on ice, and analyzed with FACSCalibur using CellQuest Pro software (version 6.0; BD Biosciences, San Jose, CA, USA).

### 2.4. Determination of FOXP3^+^ Group

ATL patient samples were analyzed at the time of submission. To minimize inter-assay variation, PBMCs from a single healthy donor (female, 45 years old) were used as controls in all the experiments. The mean proportion of FOXP3^+^ cells among CD4^+^ T cells, calculated from eight independent measurements, was set as the cutoff value (11.9%). Samples with a FOXP3^+^ frequency greater than 11.9% were defined as FOXP3^+^-high group. Reported frequencies of FOXP3^+^ cells in healthy individuals range from 5 to 15% [33,34,35].

### 2.5. Transfection and Small Interfering RNA Treatment

Small interfering RNAs (siRNAs) against FOXP3 (360253_A03, 360466_C07, and 360253_E12) and control siRNAs were purchased from Invitrogen. The siRNAs were introduced using the Neon Transfection System (Thermo Fisher SCIENTIFIC) https://www.thermofisher.cn/cn/zh/home/life-science/cell-culture/transfection/neon-electroporation-system.html, accessed on 27 August 2025. Electroporation of YT#1 and ATL-026i cells was performed at 1700 V (voltage) for 20 ms (width) at pulse 1 and at 1400 V for 20 ms at pulse 2.

### 2.6. Statistical Analysis

Welch’s *t*-test was used for the statistical analysis. Statistical significance was set at *p* < 0.05.

## 3. Results

### 3.1. Patients with ATL and an Increased Number of FOXP3^+^ Cells Exhibit Higher OX40 Expression

Abnormal proliferation of FOXP3^+^ cells was observed in approximately 60% of patients with ATL [11]. The number of FOXP3^+^ cells among CD4^+^ cells from HTLV-1-infected individuals was analyzed by flow cytometry. The frequency of FOXP3^+^ cells among CD4^+^ T cells in healthy individuals reportedly ranges from 5% to 15% [33,34,35]. According to the reference values determined in our experiments, samples with a FOXP3^+^ frequency exceeding 11.9% were defined as FOXP3^+^-high. In the FOXP3^+^-high group, FOXP3 expression was detected in 59.5% of CD4^+^ T cells, whereas in the FOXP3^+^-low group, only 2.1% of CD4 ^+^ T cells expressed FOXP3 (*p* < 0.001; Table 1, Column FOXP3^+^/CD4^+^ [%]). The mean frequency of FOXP3^+^ cells among CD4^+^ T cells in asymptomatic carriers was 6.8%, which was comparable to that of healthy individuals (Table 1, Column FOXP3^+^/CD4^+^ [%]).

To further investigate the relationship between FOXP3 and OX40 expression in ATL cells, we analyzed OX40 expression in CD25^+^ cells, a known marker of ATL cells, using flow cytometry. OX40 is expressed in a small population of resting T cells [36]. When comparing asymptomatic carriers and ATL patients, the mean proportion of OX40^+^ CD25^+^ cells in PBMCs was 2.6% as opposed to 13.1% (*p* = 0.07; Table 1, Column OX40^+^ CD25^+^/PBMCs [%]). In FOXP3^+^-high samples, 25.5% of CD25^+^ cells expressed OX40, whereas only 9.1% of CD25^+^ cells were OX40-positive in the FOXP3^+^-low group (*p* = 0.2). Notably, samples such as ATL-400, ATL-428, ATL-251, and ATL-250, which exhibited higher OX40 expression levels, consistently belonged to the FOXP3^+^-high group (Table 1, column OX40^+^/CD25^+^ [%]). These findings suggested a possible relationship between FOXP3 and OX40 expression.

### 3.2. FOXP3 Expression Associates with OX40 but Not OX40L in PBMCs Cultured for 1 Day from Patients with ATL

The expression of HTLV-1 antigens remains low in vivo; however, it is upregulated following overnight ex vivo culture [37,38,39]. Notably, the expression of OX40 and its ligand OX40L have been reported to be induced by the viral oncogenic protein Tax [22,24,25]. To elucidate the relationship between the expression of FOXP3, OX40, OX40L, and Tax in ATL cells, their expression levels were analyzed using flow cytometry. PBMCs from patients with acute-type ATL (Table 2) were cultured overnight to induce Tax antigen expression. In PBMCs from three patients with ATL, an association was observed between FOXP3 and OX40 expression, but no association was observed with OX40L expression (Figure 1A,B). The Tax expression level varied among individual patients, and in ATL-145 cells, Tax expression remained low even after overnight culture (Figure 1C). Across all three samples, some FOXP3^+^ cells expressed Tax, suggesting a possible relationship between Tax expression and FOXP3/OX40 expression (Figure 1C). These results indicate that OX40 is expressed in FOXP3^+^ cells, whereas OX40L is expressed in FOXP3^−^ cells.

### 3.3. FOXP3 Suppresses the Expression of OX40L in HTLV-1-Infected Cells

We further analyzed whether FOXP3 regulates the expression of OX40L, OX40, and Tax. Two HTLV-1-infected FOXP3-expressing cell lines, YT#1 derived from purified Treg cells from a healthy donor, and ATL-026i, derived from a patient with acute ATL, were used. FOXP3 expression in these cells was knocked down using RNA interference, and the knockdown efficiency of each siRNA was evaluated by flow cytometry (Figure 2A,B). Among the three siRNA for FOXP3, siFOXP3_E12 effectively decreased FOXP3 expression in both YT#1 and ATL-026i cells (Figure 2A,B). FOXP3 knockdown using siFOXP3_E12 increased the OX40L^+^ OX40^+^ cell population from 2.3% to 23.9% in YT#1 cells, and from 5.6% to 14.0% in ATL-026i cells (Figure 2D,F). Meanwhile, the effects of FOXP3 knockdown on OX40 and Tax expression varied depending on the cell line (Figure 2C–F). These results indicate that FOXP3 is involved in the expression of OX40L.

## 4. Discussion

Approximately 60% of patients with ATL show FOXP3^+^ ATL cells in the PB. Although a selective increase in FOXP3^+^ cells in ATL cells from patients has been previously reported [11], the mechanisms and importance of this phenomenon remain largely unknown. In this study, we found that FOXP3 expression was associated with OX40 expression in ATL cells, whereas FOXP3 reduced the expression of OX40L. These results suggest that FOXP3 is involved in the selective proliferation of FOXP3^+^ ATL cells through regulation of the OX40L/OX40 system. Stimulation with soluble OX40L has been reported to promote the expansion of mouse Foxp3^+^ Tregs, which constitutively express OX40 [26]. These findings suggest that activation of the OX40/OX40L axis in HTLV-1-infected cells expressing OX40 may be sufficient to promote the expansion of FOXP3^+^ cells.

In primary ATL cells, some samples exhibited elevated OX40 expression regardless of FOXP3 expression levels (Table 1). ATL cells are thought to accumulate mutations in the genes involved in NF-κB activation [5], rendering this pathway more readily activated. Because OX40 gene expression is regulated by the NF-κB pathway, the increased expression of OX40 in ATL cells is likely mediated through this pathway.

This study demonstrated that FOXP3 and OX40 were co-expressed in PBMCs from ATL patients following overnight culture with IL-2. When PBMCs from a healthy donor were cultured for 18 h in the presence of IL-2, we observed a slight increase in FOXP3^+^ Tregs, which express CD25, and a small fraction of these cells expressed OX40. The ATL cells expressed CD25 (Table 1). We consider that overnight culturing of ATL samples may contribute to the increase in FOXP3^+^ and OX40^+^ populations in response to IL-2 stimulation. The relationship between OX40 and FOXP3 during IL-2 stimulation appears to be closely linked to CD25 expression.

Under normal conditions, OX40L is primarily expressed by antigen-presenting cells (APCs), including activated dendritic cells (DCs), B cells, and macrophages, and plays a crucial role in the co-stimulation of naïve CD4^+^ T cells. Ex vivo experiments have shown that OX40L stimulation promotes the proliferation of human CD4^+^ T cells activated by mitogenic agents such as phorbol myristate acetate (PMA), phytohemagglutinin (PHA), or soluble anti-CD3 antibody [20]. Additionally, OX40L expression has been observed in a subset of activated T cells [40]. The interaction between OX40 and OX40L on T cells has been reported to contribute to the long-term survival of CD4^+^ T cells and enhances cytokine production [41]. In ATL patient samples, FOXP3^−^ OX40L^+^ ATL cells and FOXP3^+^ OX40^+^ ATL cells were observed (Figure 1). FOXP3 possibly gives rise to FOXP3^+^ OX40^+^ ATL cells by suppressing OX40L expression. These findings suggest that OX40/OX40L interactions may contribute to the maintenance of ATL.

Beyond its role in promoting cell survival, the OX40/OX40L system has been implicated in mediating direct cell-to-cell adhesion. For instance, in patients with ATL, OX40 expressed on leukemic cells facilitates their adhesion to vascular endothelial cells, which express OX40L [14]. Lymphocytes selectively migrate from the bloodstream into the lymph nodes via high endothelial venules. Therefore, ATL cells may also travel to the lymph nodes through OX40/OX40L-mediated interactions.

In addition, we recently reported that leukemic cells in patients with acute-type ATL proliferate in the lymph nodes rather than in the PB [42]. Since a diverse population of OX40L-expressing APCs resides in the lymph nodes, HTLV-1-infected cells expressing OX40 are likely to receive additional costimulatory signals from these APCs. Therefore, OX40/OX40L interactions within the lymphoid microenvironment may contribute to the survival and proliferation of ATL cells. Future studies should examine the expression of OX40L in vascular endothelial cells and activated dendritic cells in the lymph nodes of HTLV-1-infected individuals.

The oncogenic protein Tax has been reported to induce the expression of OX40L and OX40 [22,24,25]. However, in PBMCs cultured with IL-2, OX40, and OX40L expression was observed, even in ATL-145 cells, in which Tax expression was almost absent (Figure 1). Similarly, in ATL-072, although only a minor fraction of FOXP3^+^ cells expressed Tax, OX40 and OX40L expression was elevated (Figure 1). These findings indicate that OX40 and OX40L expression occurred independently of Tax-mediated transcriptional regulation. Notably, OX40 and OX40L expressions are controlled by NF-κB signaling, and ATL cells are known to accumulate mutations in genes involved in NF-κB activation [5]. Therefore, IL-2 stimulation is likely sufficient to activate downstream signaling in these cells, resulting in the increased expression of OX40 and OX40L.

Owing to the immunogenic nature of Tax, its expression is typically absent in freshly isolated ATL cells [37,38,39]. Little is known about the precise timing and levels of Tax expression in HTLV-1-infected individuals. In HTLV-1-infected cell lines, sporadic expression of Tax is thought to contribute to the maintenance of the infected cell population [43,44]. Tax expression is induced through activation of the p38 mitogen-activated protein kinase (MAPK) signaling pathway, which itself is triggered by various environmental stresses and inflammatory cytokines [45]. Moreover, Tax can activate the p38-MAPK pathway [46], thereby establishing a positive feedback loop. Thus, Tax expression in HTLV-1-infected individuals is expected to be induced by stress signals or cytokine stimulation, and the Tax-mediated upregulation of OX40L and OX40 may enhance OX40L/OX40 interactions.

The analysis of FOXP3-expressing cells in humans is complex, since conventional human effector T cells transiently express FOXP3 and other Treg surface markers upon TCR stimulation and do not possess the functional characteristics of Treg cells. The stability of Treg cell differentiation is ensured by the DNA demethylation status of the Treg-specific demethylated region (TSDR) in the FOXP3 locus rather than by FOXP3 expression itself [47,48,49]. DNA methylation in the promoter region suppresses gene expression by preventing the binding of transcription factors. In naturally occurring tTreg and pTreg cells, which constitutively express FOXP3, the CpG islands of TSDR are demethylated [50,51]. In contrast, iTregs and conventional T cells with low or no FOXP3 expression are methylated in the TSDR region [52,53]. The methylation status of the TSDR is used to distinguish between naturally occurring Treg cells, iTreg cells, and conventional T cells. Methylation analysis of the TSDR in ATL cells revealed that approximately 58% of patients with ATL had hypomethylated TSDR, and these cells expressed FOXP3 [13]. These results support the previous findings showing that 60% of patients with ATL had FOXP3^+^ cells [11]. In addition, ATL cells from patients with hypomethylated TSDR exhibited immunosuppressive function, whereas ATL cells with a methylated TSDR did not [13]. Since HTLV-1 infects CD4^+^ T cells and induces oncogenesis, ATL cells may abnormally proliferate, originating from both naturally occurring Tregs and effector T cells.

In this study, we compared the proportions of OX40^+^ CD25^+^ cells in PBMCs between asymptomatic carriers (n = 10) and patients with ATL (n = 14). Although the proportion tended to be higher in patients with ATL, the difference did not reach statistical significance (2.6% vs. 13.1%; *p* = 0.07; Table 1, column OX40^+^ CD25^+^/PBMCs [%]). Similarly, when ATL cells were classified into FOXP3^+^-high (n = 9) and FOXP3^+^-low (n = 5) groups, the proportion of OX40-expressing cells showed an increasing trend in the FOXP3^+^-high group, but the difference was not statistically significant (25.5% vs. 9.1%; *p* = 0.2; Table 1, column OX40^+^/CD25^+^ [%]). One limitation of this study is the small number of available samples. Because individual ATL cells exhibit highly diverse gene expression profiles, establishing statistical significance requires analysis of a larger cohort of ATL patient samples. To demonstrate a definitive and statistically significant association between FOXP3 and OX40 expression in ATL cells, further studies with expanded sample sizes are warranted.

In conclusion, this study demonstrated that FOXP3 is involved in the downregulation of OX40L in HTLV-1-infected cells. Given the positive association between FOXP3 and OX40 expression in ATL cell populations, these findings suggest that interactions between coexisting FOXP3^−^ OX40L^+^ cells and FOXP3^+^ OX40^+^ ATL cells may promote the proliferation and survival of FOXP3^+^ ATL cells. Collectively, these findings imply that the OX40/OX40L axis may facilitate the expansion of FOXP3^+^ cells through reciprocal stimulation within the lymph nodes.

## Figures and Tables

**Figure 1 viruses-17-01445-f001:**
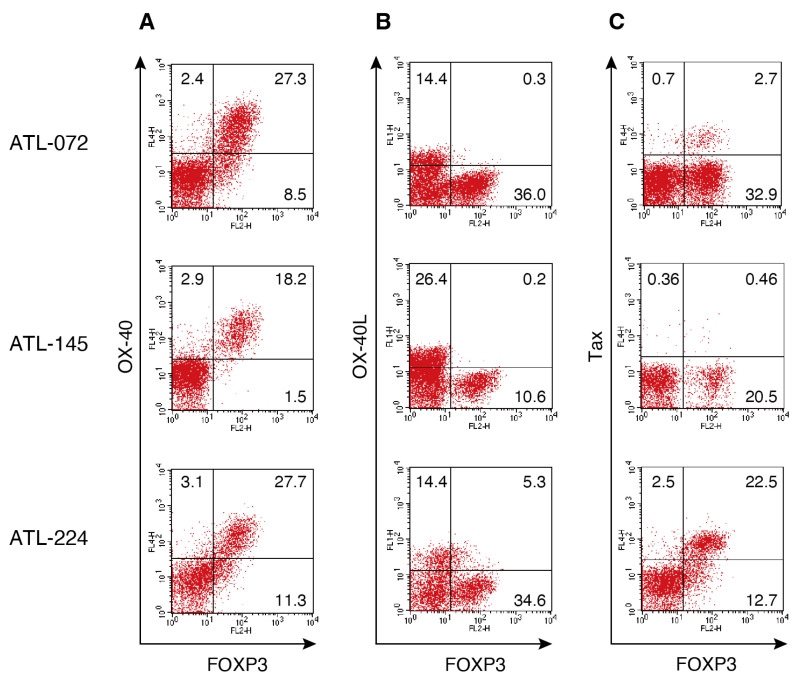
Association between FOXP3 and OX40 expression in ATL cells. (**A**–**C**) Freshly isolated PBMCs from patients with ATL were cultured for 18 h in the presence of IL-2. The cells were stained with HyLite Fluor 647-conjugated anti-OX40, FITC-conjugated anti-OX40L, PE-conjugated anti-FOXP3, and HyLite Fluor 647-conjugated anti-Tax antibodies. Representative flow cytometry plots gated on CD4^+^ T cells are shown. Three independent experiments were performed once, each using PBMCs derived from three patients with the acute type of ATL.

**Figure 2 viruses-17-01445-f002:**
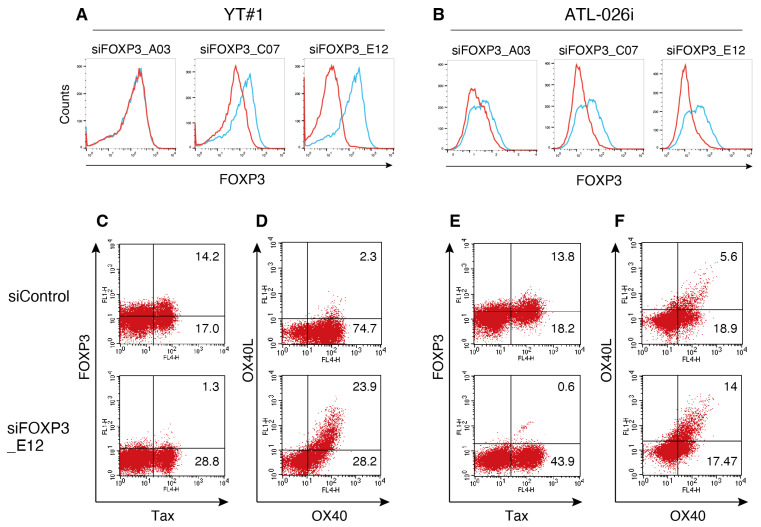
FOXP3-mediated suppression of OX40L in HTLV-1-infected cell lines. (**A**,**B**) YT#1 and ATL-026i cells were transfected with FOXP3-specific siRNA and cultured for 48 h. The cells were stained with PE-conjugated anti-FOXP3 antibody and analyzed by flow cytometry. The red and blue lines indicate siFOXP3- and siControl-treated cells, respectively. (**C**–**F**) YT#1 and ATL-026i cells transfected with siFOXP3_E12 were cultured for 48 h and stained with HyLite Fluor 647-conjugated anti-OX40, FITC-conjugated anti-OX40L, FITC-conjugated anti-FOXP3, and HyLite Fluor 647-conjugated anti-Tax antibodies. Representative flow cytometry plots are shown. Two independent experiments were performed once each using FOXP3-expressing cell lines.

**Table 1 viruses-17-01445-t001:** Characterization of FOXP3^+^ and OX40^+^ cell populations in freshly isolated PBMCs from HTLV-1-infected individuals. The clinical characteristics of HTLV-1-infected individuals are listed. Freshly isolated PBMCs from HTLV-1-infected individuals were stained with FITC-conjugated anti-FOXP3 and PE/cyanine7-conjugated anti-CD4 antibodies, and FOXP3^+^ CD4^+^ T cells were quantified by gating the CD4^+^ population. The cells were stained with FITC-conjugated anti-CD25 and HyLite Fluor 647-conjugated anti-OX40 antibodies, and CD25^+^ OX40^+^ T cells were quantified by gating the CD25^+^ population.

	Group	Bank Number	Diagnosis	Sex	Age	FOXP3^+^/ CD4^+^ (%)	OX40^+^ CD25^+^/ PBMCs (%)	OX40^−^CD25^+^/ PBMCs (%)	Total CD25^+^/ PBMCs (%)	OX40^+^ /CD25^+^ (%)
1	FOXP3^+^ high	ATL-420	Acute	M	58	85.61	9.95	54.6	64.55	15.42
2		ATL-424	Acute	M	87	82.58	4.9	76.09	80.99	6.06
3		ATL-400	Acute	F	71	76.19	59.57	12.43	72	82.74
4		ATL-428	Chronic	M	79	70.49	45.95	48.53	94.48	48.64
5		ATL-250	Chronic	M	60	65.38	17.01	29.96	46.97	36.22
6		ATL-408	Acute	F	79	49.76	2.17	77.91	80.08	2.71
7		ATL-414	Acute	F	80	43.09	2.68	80.86	83.54	3.21
8		ATL-251	Acute	F	64	39.37	10.78	25.22	36	29.95
9		ATL-388	Acute	M	61	22.94	2.64	52.5	55.14	4.79
10	FOXP^+^ low	ATL-421	Acute	F	53	8.11	2.33	24.23	26.56	8.78
11		ATL-412	Acute	M	39	1.05	5.22	41.9	47.12	11.08
12		ATL-401	Acute	M	68	0.83	13.58	67.53	81.11	16.75
13		ATL-402	Acute	M	70	0.72	5.58	73.28	78.86	7.08
14		ATL-405	Acute	M	66	0	0.9	48.8	49.7	1.82
15	Carrier	ATL-427	Carrier	F	90	12.2	3.08	9.76	12.84	
16		ATL-399	Carrier	F	63	11.82	2.48	5.88	8.36	
17		ATL-410	Carrier	M	79	9.81	1.24	14.75	15.99	
18		ATL-418	Carrier	M	67	8.41	3.99	15.05	19.04	
19		ATL-422	Carrier	F	67	8.28	3.07	12.52	15.59	
20		ATL-423	Carrier	F	58	5.2	1.84	8.89	10.73	
21		ATL-426	Carrier	M	54	4.08	2.54	20.92	23.46	
22		ATL-419	Carrier	F	79	3.95	2.13	10.66	12.79	
23		ATL-413	Carrier	M	44	3.04	2.78	39.07	41.85	
24		ATL-417	Carrier	F	87	1.59	2.8	20.29	23.09	

**Table 2 viruses-17-01445-t002:** Clinical characteristics of HTLV-1-infected individuals.

	Bank Number	Diagnosis	Sex	Age
1	ATL-072	Acute	F	75
2	ATL-145	Acute	F	64
3	ATL-224	Acute	M	80

## Data Availability

The original contributions presented in this study are included in the article. Further inquiries can be directed to the corresponding authors.

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
