# Peer review of "Association Between FOXP3 and OX40 Expression in Adult T-Cell Leukemia Cells"

_viruses, 2025, doi:10.3390/v17111445_

Round 1
Reviewer 1 Report
Comments and Suggestions for Authors
This manuscript reports the results of a study focused on the possible role of the interplay among FOXP3+, OX40, OX40L in the proliferation of ATL cells. OX40 expression in T cells was more elevated in a restricted number of ATL patients with high frequency of FOXP3⁺/CD4+ cells with respect to a group of ATL patients with low frequency of FOXP3⁺/CD4+ cells. In 18 h cultured PBMCs from 3 patients with acute ATL, FOXP3⁻ cells mainly expressed OX40L but not OX40, while FOXP3⁺ cells mainly expressed OX40 cells but not OX40L. In addition, siRNA-mediated FOXP3 knockdown in HTLV-1-infected cell lines increased OX40L expression. Authors conclude that the study demonstrates that FOXP3 is involved in the downregulation of OX40L in HTLV-1-infected cells and that the correlation FOXP3/OX40 expression in ATL cells suggest that interactions between FOXP3⁻ OX40L⁺ cells and FOXP3⁺ OX40⁺ cells may promote the proliferation of FOXP3⁺ ATL cells.
In the opinion of this reviewer, the subject of the study is of potential interest. However, a logical interpretation of the results is difficult in the absence of additional data which could contribute to better understand the significance of this study.
Suggested additional data and changes.
Main concern
1) The term “correlation” in scientific studies properly refers to a relationship between two or more variable that can be analyzed and expressed as a statistical measure. In medical/biological studies the relationship among variables must be statistically analyzed by standard methods such as the calculation of the Pearson’s correlation coefficient. Thus, it is surprising that a manuscript which is entitled “Correlation between FOXP3 and OX4….” does not contain any specific statistical analysis that can support this affirmation! Authors should: i) revise their manuscript by providing such a kind of statistical analysis to support the noun “correlation” or the verb “to correlate” all over the text where these words are referred to their results, ii) alternatively, explain why they did not include these analyses in their manuscript (did they make them with negative results?). In any case, over all of the manuscript, including the title, the words “correlation, correlates” should be excluded when referred to the results of this study and substituted with other words (relationship? association?), not to confuse the reader. In addition, keeping in mind that in any case correlation, even when statistically supported, never automatically implies a causative link among variables, authors should be very cautious about the explanations of the results their obtained and the conclusions of their study. The manuscript should be attentively rewritten based on this main concern.
Other concerns
2) Table 1 reports data from isolated PBMCs from 14 ATL patients. It could be interesting to compare data from ATL patients with data from age-matched healthy donors to verify how specific are the observed results for ATL patients.
3) Figure 1 is not supported by any statistical data. Please provide them.
4) Figure 2, differently from Table 1 and Figure 1 that report data from clinical samples, refers to an experimental phase of the study. However, no information is given on experimental design of this phase of the study and on the reproducibility of the experiment, such as how many times the experiment was repeated with similar results. Again, results reported in Figure 2 are not supported by a basic statistical analysis, such as means and S.D. obtained from independent experiments. Please provide them.
Minor
5) in Table 2, the two groups of high and low FOXP3 expressing cells are designed as FOXP3 hyphen high/hyphen low, where hyphen may be confused with minus. Probably, better FOXP3+ high, FOXP3+ low.
Author Response
Responses to Reviewer #1
Comment #1-1
The term “correlation” in scientific studies properly refers to a relationship between two or more variable that can be analyzed and expressed as a statistical measure. In medical/biological studies, the relationship among variables must be statistically analyzed by standard methods, such as the calculation of the Pearson’s correlation coefficient. Thus, it is surprising that a manuscript which is entitled “Correlation between FOXP3 and OX4….” does not contain any specific statistical analysis that can support this affirmation! Authors should: i) revise their manuscript by providing such a kind of statistical analysis to support the noun “correlation” or the verb “to correlate” all over the text where these words are referred to their results, ii) alternatively, explain why they did not include these analyses in their manuscript (did they make them with negative results?). In any case, over all of the manuscript, including the title, the words “correlation, correlates” should be excluded when referred to the results of this study and substituted with other words (relationship? association?), not to confuse the reader. In addition, keeping in mind that in any case correlation, even when statistically supported, never automatically implies a causative link among variables, authors should be very cautious about the explanations of the results their obtained and the conclusions of their study. The manuscript should be attentively rewritten based on this main concern.
Response #1-1
Thank you for your comments. To convey the content of this manuscript accurately, we have carefully revised the text as suggested.
- We replaced the expressions “correlation” and “correlate” in descriptions of experimental results where statistical analyses could not be included, with the terms “relationship” or “association” in Title, Results section 3.2. and Figure 1.
- Statistical analyses could not be included in these parts because patient samples were rare and difficult to obtain, making it challenging to perform the same experiment three times. To compensate for this, experiments were conducted using samples from three independent patients. The flow cytometry results demonstrated that FOXP3-positive cells expressed OX40, whereas FOXP3-negative cells expressed OX40L in these three independent samples. As stated in the Figure 1 legend, “Three independent experiments were performed once, each using PBMCs derived from three patients with the acute type of ATL.” in lines 359–360.
- In the Discussion section, the portions where we also changed “correlation” to “association” correspond to the explanation of the results shown in Results section 3.2. and Figure 1.
Comment #1-2
Table 1 reports data from isolated PBMCs from 14 ATL patients. It could be interesting to compare data from ATL patients with data from age-matched healthy donors to verify how specific are the observed results for ATL patients.
Response #1-2
Because ATL typically develops later in life, obtaining age-matched healthy donors is challenging. However, as we had data from asymptomatic carriers, we have incorporated these results into Table 1. Specifically, we included the frequencies of FOXP3⁺ cells among CD4⁺ T cells, as well as OX40⁺ CD25⁺ cells, OX40⁻ CD25⁺ cells, and total CD25⁺ cells within PBMCs from HTLV-1–infected individuals.
Based on these results, we added the following statements to the text:
- “The mean frequency of FOXP3⁺ cells among CD4⁺ T cells in asymptomatic carriers was 6.8%, which was comparable to that of healthy individuals (Table 1, Column FOXP3⁺/CD4⁺ [%]).” (page 4, lines 205–207).
- “When comparing asymptomatic carriers and ATL patients, the mean proportion of OX40⁺ CD25⁺ cells in PBMCs was 2.6% as opposed to 13.1% (p = 0.07; Table 1, Column OX40⁺ CD25⁺/PBMCs [%]).” (page 4, lines 210–213).
Comment #1-3
Figure 1 is not supported by any statistical data. Please provide them.
Response #1-3
Because samples from HTLV-1–infected individuals are rare and difficult to obtain, performing the same experiment three times was challenging. To address this limitation, we performed an experiment using samples from three independent patients. We demonstrated that FOXP3-positive cells expressed OX40, whereas FOXP3-negative cells expressed OX40L in the three independent samples. We have stated that “Three independent experiments were performed once, each using PBMCs derived from three patients with the acute type.” in Figure 1 legend, lines 359–360.
Comment #1-4
Figure 2, differently from Table 1 and Figure 1 that report data from clinical samples, refers to an experimental phase of the study. However, no information is given on experimental design of this phase of the study and on the reproducibility of the experiment, such as how many times the experiment was repeated with similar results. Again, results reported in Figure 2 are not supported by a basic statistical analysis, such as means and S.D. obtained from independent experiments. Please provide them.
Response #1-4
Using two independent FOXP3-positive cell lines, we confirmed that FOXP3 knockdown led to increased OX40L expression. We described the sentence “Two independent experiments were performed once each using FOXP3-expressing cell lines.” of Figure 2 (lines 382–383). Owing to changes in the experimenter’s affiliation, it has become difficult to repeat the experiments. Further statistical significance testing is required to fully assess the reproducibility of the experiments and the trends indicated by the representative data, and additional experiments will be necessary in the future.
Comment #1-5
In Table 1, the two groups of high and low FOXP3 expressing cells are designed as FOXP3 hyphen high/hyphen low, where hyphen may be confused with minus. Probably, better FOXP3+ high , FOXP3+ low.
Response #1-5
Thank you for your comments. We have revised Table 1.

Reviewer 2 Report
Comments and Suggestions for Authors
In the manuscript, the authors wished to investigate the mechanisms driving
the expansion of FOXP3⁺ in ATL cells. The use of primary ATL patients is commended in this manuscript; however, the limited number prevents strong statistical correlations to be inferred.
Although some ATL patients with high OX40 expression fall within the FoxP3⁺ group, there does not appear to be a consistent correlation. Similarly, patients with high OX40 expression are also observed in the FoxP3⁻ group, and vice versa. The authors should perform statistical correlation analyses to validate these observations. In addition, how do the authors explain ATL patients who display high FoxP3 expression but low OX40 levels?
For figure 1, it would have been good to know what the original levels of FoxP3, OX40, and OX40L were, prior to overnight stimulation with IL-2, as presented in Table 1. Does growth in IL-2 affect the levels? Does overnight culturing of these ex vivo ATL samples disrupt the FoxP3/OX40 correlation seen in Table 1?
The statement, “Tax expression does not correlate with FOXP3 expression” cannot be inferred with statistical significance from figure 1, where only 3 ATL patients are used. In Figure 1, ATL145 FoxP3+ cells express the least amount of OX40 with undetectable Tax, whereas ATL 072 and ATL 224 have measurable Tax and the most OX40 expression. If FoxP3 and OX40 correlate (from Table 1), this would suggest that Tax correlate with FoxP3/OX40 expression.
In ATL patients expressing low FoxP3 (Table 1), does ex vivo culturing with soluble OX40, show enhanced FoxP3 expression?
For Figure 2A: It would be good to show RNA or protein loss of FoxP3 after siRNA.
In the first paragraph (lines 132-139), the authors should make it clear what cells they are analyzing – it is not directly clear whether they are talking about PBMCs or ATL at first read.
Comments on the Quality of English LanguageGood
Author Response
Responses to Reviewer #2
Comment #2-1
In the manuscript, the authors wished to investigate the mechanisms driving the expansion of FOXP3⁺ in ATL cells. The use of primary ATL patients is commended in this manuscript; however, the limited number prevents strong statistical correlations to be inferred.
Although some ATL patients with high OX40 expression fall within the FoxP3⁺ group, there does not appear to be a consistent correlation. Similarly, patients with high OX40 expression are also observed in the FoxP3⁻ group, and vice versa. The authors should perform statistical correlation analyses to validate these observations. In addition, how do the authors explain ATL patients who display high FoxP3 expression but low OX40 levels?
Response #2-1
Thank you for your comments.
We performed statistical analysis of OX40⁺/CD25 (%) in the FOXP3⁺ high group and the FOXP3⁺ low group (p = 0.2) (page 4, line 214). Since no statistical significance was observed, we described the concluding statement as “These findings suggested a relationship between FOXP3 and OX40 expression.”
The expression of OX40 is regulated by NF-κB and is not under the control of FOXP3 as shown in Figure 2. We considered that specimens with high FOXP3 expression represented ATL cells derived from the expansion of naturally occurring Tregs, whereas specimens with low FOXP3 expression represented ATL cells derived from the expansion of effector T cells. We previously reported that ATL cells in the peripheral blood do not proliferate actively (Mizuguchi et al., Cancer Gene Therapy, 2022). Therefore, ATL patient samples with high FOXP3 expression and low OX40 levels likely reflect a state in which ATL cells are not activated at the time of collection.
Comment #2-2
For figure 1, it would have been good to know what the original levels of FoxP3, OX40, and OX40L were, prior to overnight stimulation with IL-2, as presented in Table 1. Does growth in IL-2 affect the levels? Does overnight culturing of these ex vivo ATL samples disrupt the FoxP3/OX40 correlation seen in Table 1?
Response #2-2
We also agree that comparing the baseline levels of FOXP3, OX40, and OX40L prior to overnight stimulation with IL-2 would be valuable. However, we did not collect these data at the time of sample acquisition and no longer had those samples available. In future studies, we plan to obtain data before and after stimulation using newly acquired samples.
When PBMCs from a healthy donor were cultured for 18 h in the presence of IL-2, we observed a slight increase in CD25-expressing FOXP3+ Tregs, and a small fraction of these cells expressed OX40 (data not shown). ATL cells expressed CD25 (Table 1). We consider that overnight culturing of ATL samples may contribute to the increase in FOXP3⁺ and OX40⁺ populations in response to IL-2 stimulation, and that this would not affect the observed associations.
Comment #2-3
The statement, “Tax expression does not correlate with FOXP3 expression” cannot be inferred with statistical significance from Figure 1, where only 3 ATL patients are used. In Figure 1, ATL145 FoxP3+ cells express the least amount of OX40 with undetectable Tax, whereas ATL 072 and ATL 224 have measurable Tax and the most OX40 expression. If FoxP3 and OX40 correlate (from Table 1), this would suggest that Tax correlate with FoxP3/OX40 expression.
Response #2-3
Thank you for the suggestion. We described the sentence “Across all three samples, some FOXP3+ cells expressed Tax, suggesting a possible relationship between Tax expression and FOXP3/OX40 expression (Figure 1C).” on page 5, lines 337–338, in Results section 3.2. section, and deleted the sentence “Tax expression does not correlate with FOXP3 expression.”
Comment #2-4
In ATL patients expressing low FoxP3 (Table 1), does ex vivo culturing with soluble OX40 show enhanced FoxP3 expression?
Response #2-4
We believe that samples with low FOXP3 expression are those in which effector T cells have expanded. Because FOXP3 expression in effector T cells requires IL-2 and TGF-β stimulation in mouse model (Mikami et al., PNAS, 2020), we speculate that stimulation with OX40 alone may not enhance FOXP3 expression.
Comment #2-5
For Figure 2A: It would be good to show RNA or protein loss of FoxP3 after siRNA.
Response #2-5
Figure 2A and B show that introducing various siRNAs targeting FOXP3 into HTLV-1–infected cell lines reduces FOXP3 expression at the intracellular protein level.
Comment #2-6
In the first paragraph (lines 132-139), the authors should make it clear what cells they are analyzing; it is not directly clear whether they are talking about PBMCs or ATL at first read.
Response #2-6
We have revised the manuscript to clarify that ATL cells are the focus of our analysis on page 4, lines 199–200, in Results section 3.1. section.

Reviewer 3 Report
Comments and Suggestions for Authors
The retrovirus HTLV-1 is the cause of the aggressive malignancy ATL. A majority of ATL cells express FOXP3 and these HTLV-1-infected Treg cells have an immune-suppressive function. Aberrant expansion of FOXP3+ cells is thought to be involved in development of ATL. This manuscript investigates the mechanism underlying expansion of FOXP3+ ATL cells. OX40 is a costimulatory molecule and is expressed in ATL cells. Using ATL patient samples and flow cytometry, the authors examined expression of OX40 and its ligand OX40L. By examining a limited subset of ATL patient cells, FOXP3+ cells express OX40 while FOXP3- cells express OX40L. Using siRNA to knockdown FOXP3, the authors demonstrate FOXP3 regulates expression of OX40L. Taken together, these results suggest FOXP3 contributes to proliferation of FOXP3+ ATL cells by regulating the OX40/OX40L system. The manuscript is well written and results are clearly presented. One concern is noted below:
Figure 1: Based on the authors presented findings, the viral sense gene Tax does not corelate with FOXP3 expression. Given its consistent expression throughout infection and disease and its reported activation of FOXP3, does the anti-sense viral gene Hbz correlate with FOXP3 expression or OX40/OX40L? The authors should also include the previous literature describing Hbz activation of FOXP3 in the Introduction.
Author Response
Responses to Reviewer #3
Comment #3-1
Figure 1: Based on the authors presented findings, the viral sense gene Tax does not corelate with FOXP3 expression. Given its consistent expression throughout infection and disease and its reported activation of FOXP3, does the anti-sense viral gene Hbz correlate with FOXP3 expression or OX40/OX40L? The authors should also include the previous literature describing Hbz activation of FOXP3 in the Introduction.
Response #3-1
Thank you for your comments. Regarding whether the anti-sense viral gene HBZ correlates with FOXP3 expression or with OX40/OX40L, we have not yet analyzed HBZ expression together with OX40/OX40L expression. In the future, we plan to establish an experimental system to investigate this issue.
We have included in the Introduction a reference to previous studies describing HBZ-mediated expansion of Foxp3+ T cells on page 2, lines 55–56, adding the following sentence: “Transgenic mice expressing the antisense viral gene HTLV-1 bZIP factor (HBZ) in CD4+ T cells develop lymphomas expressing Foxp3 [12].”

Reviewer 4 Report
Comments and Suggestions for Authors
The study by Mizuguchi et al. examined FOXP3 expression and its interactions with OX40, OX40L, and Tax in ATL cells. The authors investigated the role of FOXP3 in ATL cell development and proliferation and the presented data suggested a positive correlation between FOXP3 and OX40 expression. While the topic is interesting and relevant to the journal’s scope, several critical concerns limit the clarity and rigor of the manuscript. As such, the manuscript appears premature in its current form and would benefit from either additional data and analysis. In conclusion, the perspective of the article is interesting, but the paper needs improvement to be recommended for publication.
Major Points
1. Definition of FOXP3-high and control group (Lines 135–136)
The manuscript states that samples with FOXP3⁺ frequencies exceeding 11.9% were classified as FOXP3-high, based on healthy control values. No information is provided about the healthy controls, such as sample size, age distribution, or selection criteria. This omission undermines the justification for defining the 11.9% cutoff. Authors should provide detailed information on the control group and justify its comparability with ATL patient samples.
2. OX40 expression in resting T cells (Line 142)
The authors claim that OX40 is not expressed in resting T cells. Please cite primary literature demonstrating the absence of OX40 in resting T cells.
3. Correlation of FOXP3 and OX40 after IL-2 stimulation (Lines 163–167)
The study reports a correlation between FOXP3 and OX40 in PBMCs from ATL patients cultured briefly with IL-2. It is unclear whether this correlation is unique to ATL cells or could also occur in IL-2–stimulated PBMCs from healthy donors. Please clarify whether the observation is ATL-specific and discuss its biological significance with appropriate references.
4. FOXP3 knockdown and OX40L/OX40 expression (Lines 188–189)
The authors report that FOXP3 knockdown using siFOXP3_E12 increased the OX40L⁺ population in YT#1 and ATL-026i cells. The manuscript does not report the number of independent trials or statistical analysis supporting the conclusion that FOXP3 knockdown enhances OX40L expression. Authors should provide the number of replicates, statistical significance, and error bars to substantiate this conclusion.
5. Interactions between FOXP3⁻ OX40L⁺ and FOXP3⁺ OX40⁺ ATL cells (Lines 268–272)
The authors suggest that these interactions may promote proliferation and survival of FOXP3⁺ ATL cells. The data show that FOXP3 downregulation does not affect OX40 expression, and Tax expression does not correlate with OX40. Moreover, OX40L appears downregulated in FOXP3⁺ ATL cells. Authors should discuss potential mechanisms by which ATL cells maintain high OX40 expression despite FOXP3 levels. Also, elaborate on how downregulation of OX40L in FOXP3⁺ cells might impact survival or proliferation of ATL cells.
6. Correlation with Tax expression
Previous studies reported that Tax can upregulate OX40 and OX40L. The manuscript does not clarify whether a correlation between Tax and these molecules was observed in the authors’ experiments. Authors should provide an analysis of Tax expression in the context of OX40 and OX40L expression and discuss any discrepancies with previous reports.
Comments on the Quality of English LanguageThe English is largely fine, but The could be improved to more clearly express the research.
Author Response
Responses to Reviewer #4
Comment #4-1: Definition of FOXP3-high and control group (Lines 135–136)
The manuscript states that samples with FOXP3⁺ frequencies exceeding 11.9% were classified as FOXP3-high, based on healthy control values. No information is provided about the healthy controls, such as sample size, age distribution, or selection criteria. This omission undermines the justification for defining the 11.9% cutoff. Authors should provide detailed information on the control group and justify its comparability with ATL patient samples.
Response #4-1
Thank you for your comment.
We have added the following description regarding the healthy donor and the method for determining the cutoff value in the Materials and Methods section on page 3, lines 129–135.
2.4. Determination of FOXP3⁺ Group
ATL patient samples were analyzed at the time of submission. To minimize inter-assay variation, PBMCs from a single healthy donor (female, 45 years old) were used as controls in all the experiments. The mean proportion of FOXP3⁺ cells among CD4⁺ T cells, calculated from eight independent measurements, was set as the cutoff value (11.9%). Samples with a FOXP3⁺ frequency greater than 11.9% were defined as the FOXP3-high group. Reported frequencies of FOXP3⁺ cells in healthy individuals range from 5 to 15% [33-35].
In addition, to determine whether high FOXP3 expression was specific to ATL, we added the results for asymptomatic carriers (Table 1). The mean FOXP3+ frequency among CD4+ T cells in carriers was 6.8%, supporting the appropriateness of the chosen cutoff value.
Comment #4-2: OX40 expression in resting T cells (line 142)
The authors claim that OX40 is not expressed in resting T cells. Please cite primary literature demonstrating the absence of OX40 in resting T cells.
Response #4-2
In peripheral blood T cells from healthy individuals, while CD8⁺ T cells showed no expression, a small population of CD4⁺ T cells expressed OX40 (Takasawa et al., Jpn. J. Cancer Res., 2001). Citing this reference, we included the following sentence on page 4, line 168: “OX40 is expressed in a small population of resting T cells [36].”
Comment #4-3: Correlation of FOXP3 and OX40 after IL-2 stimulation (Lines 163–167)
The study reports a correlation between FOXP3 and OX40 in PBMCs from ATL patients cultured briefly with IL-2. It is unclear whether this correlation is unique to ATL cells or could also occur in IL-2–stimulated PBMCs from healthy donors. Please clarify whether the observation is ATL-specific and discuss its biological significance with appropriate references.
Response #4-3
We demonstrated the relationship between FOXP3 and OX40 in PBMCs from ATL patients following overnight culture with IL-2. Although preliminary, our data showed that when PBMCs from a healthy donor was cultured in the presence of IL-2, the subpopulation of FOXP3⁺ cells expressing CD25 increased slightly, and OX40 expression was observed in a part of these cells. These findings suggest that this phenomenon is not specific to ATL cells but rather characteristic of FOXP3⁺ cells expressing CD25.
We have added the following statement to the Discussion on page 8, lines 360–367.
“This study demonstrated that FOXP3 and OX40 were co-expressed in PBMCs from ATL patients with ATL following overnight culture with IL-2. When PBMCs from a healthy donor were cultured for 18 h in the presence of IL-2, we observed a slight increase in FOXP3+ Tregs, which express CD25, and a small fraction of these cells expressed OX40 (data not shown). The ATL cells expressed CD25 (Table 1). We consider that overnight culturing of ATL samples may contribute to the increase in FOXP3⁺ and OX40⁺ populations in response to IL-2 stimulation. The relationship between OX40 and FOXP3 during IL-2 stimulation appears to be closely linked to CD25 expression.”
Comment #4-4: FOXP3 knockdown and OX40L/OX40 expression (Lines 188–189)
The authors report that FOXP3 knockdown using siFOXP3_E12 increased the OX40L⁺ population in YT#1 and ATL-026i cells. The manuscript does not report the number of independent trials or statistical analysis supporting the conclusion that FOXP3 knockdown enhances OX40L expression. Authors should provide the number of replicates, statistical significance and error bars to substantiate this conclusion.
Response #4-4
Using two independent FOXP3-positive cell lines, we confirmed that FOXP3 knockdown led to increased OX40L expression. We described the sentence “Two independent experiments were performed once each using FOXP3-expressing cell lines.” of Figure 2 (lines 340–341). Owing to changes in the experimenter’s affiliation, it has become difficult to repeat the experiments. Further statistical significance testing is required to fully assess the reproducibility of the experiments and the trends indicated by the representative data, and additional experiments will be necessary in the future.
Comment #4-5: Interactions between FOXP3⁻ OX40L⁺ and FOXP3⁺ OX40⁺ ATL cells (Lines 268–272)
The authors suggest that these interactions may promote proliferation and survival of FOXP3⁺ ATL cells. The data show that FOXP3 downregulation does not affect OX40 expression, and Tax expression does not correlate with OX40. Moreover, OX40L appears downregulated in FOXP3⁺ ATL cells. Authors should discuss potential mechanisms by which ATL cells maintain high OX40 expression despite FOXP3 levels. Also, elaborate on how downregulation of OX40L in FOXP3⁺ cells might impact survival or proliferation of ATL cells.
Response #4-5
In primary ATL cells, some samples exhibited elevated OX40 expression regardless of FOXP3 expression levels (Table 1). ATL cells are thought to accumulate mutations in genes involved in NF-κB activation (Kataoka et al., Nature Genetics, 2015), rendering this pathway more readily activated. Because OX40 gene expression is regulated by the NF-κB pathway, the increased expression of OX40 in ATL cells is likely attributable to this pathway. We have included this point in the text (pages 7-8, lines 354–359 in the Discussion section).
We discussed how the downregulation of OX40L in FOXP3⁺ cells may potentially affect the survival and proliferation of ATL cells in the Discussion section on page 8, lines 376–379, as described below. “In ATL patient samples, FOXP3⁻ OX40L⁺ ATL cells and FOXP3⁺ OX40+ ATL cells were observed (Figure 1). FOXP3 possibly gives rise to FOXP3⁺ OX40⁺ ATL cells by suppressing OX40L expression. These findings suggest that OX40/OX40L interactions may contribute to the maintenance of ATL.”
Comment #4-6: Correlation with Tax expression
Previous studies reported that Tax can upregulate OX40 and OX40L. The manuscript does not clarify whether a correlation between Tax and these molecules was observed in the authors’ experiments. Authors should provide an analysis of Tax expression in the context of OX40 and OX40L expression and discuss any discrepancies with previous reports.
Response #4-6
In the ATL samples ATL-072 and ATL-224, OX40 and OX40L expression was not regulated by Tax. We have discussed the possible mechanisms of Tax-independent gene regulation and included the following statement in the Discussion section on page 8, lines 394–403.
“The oncogenic protein Tax has been reported to induce the expression of OX40L and OX40 [22, 24, 25]. However, in PBMCs cultured with IL-2, OX40, and OX40L, Tax expression was observed, even in ATL‑145 cells, in which Tax expression was almost absent (Figure 1). Similarly, in ATL‑072, although only a minor fraction of FOXP3⁺ cells expressed Tax, OX40 and OX40L expression was elevated (Figure 1). These findings indicated that OX40 and OX40L expression occurred independently of Tax-mediated transcriptional regulation. Notably, OX40 and OX40L expression is controlled by NF-κB signaling, and ATL cells are known to accumulate mutations in genes involved in NF-κB activation [5]. Therefore, IL-2 stimulation is likely sufficient to activate downstream signaling in these cells, resulting in the increased expression of OX40 and OX40L.”
Comment #4-7
The English is largely fine, but The could be improved to more clearly express the research.
Response #4-7
Thank you. Accordingly, the manuscript has been professionally proofread. We have carefully selected the wording to ensure that the content of the paper is accurately conveyed.

Round 2
Reviewer 1 Report
Comments and Suggestions for Authors
All of the points raised in my comments have been sufficiently addressed.
Author Response
Comments #1-1
All of the points raised in my comments have been sufficiently addressed.
Response #1-1
We sincerely appreciate the reviewer’s constructive comments.
Reviewer 2 Report
Comments and Suggestions for Authors
While the authors addressed the reviewers’ comments in writing, no corresponding changes were made to the actual experiments. The manuscript does not read as a complete study, but rather as a preliminary observation that requires further experimental validation. The lack of robust statistical analysis, the limited number of patient samples, and the absence of strong correlations detract from the overall impact of this manuscript. The authors present an interesting observation; however, it is not fully supported by comprehensive experimental validation. As currently written, it feels incomplete and lacks sufficient evidence to justify publication.
Author Response
Comment #2-1
While the authors addressed the reviewers’ comments in writing, no corresponding changes were made to the actual experiments. The manuscript does not read as a complete study, but rather as a preliminary observation that requires further experimental validation. The lack of robust statistical analysis, the limited number of patient samples, and the absence of strong correlations detract from the overall impact of this manuscript. The authors present an interesting observation; however, it is not fully supported by comprehensive experimental validation. As currently written, it feels incomplete and lacks sufficient evidence to justify publication.
Response #2-1
Thank you again for your helpful comments. We agree that the present findings need to be supported by additional large-scale experiments, as each ATL patient sample is unique and exhibits diverse gene expression patterns and variations.
In the Results section 3.1, to convey the findings accurately, we used the term “relationship” instead of “correlation,” which implies statistical significance, and summarized the results as follows:
“These findings suggested a possible relationship between FOXP3 and OX40 expression” (page 4, lines 160–161).
In addition, we believe that the title of the paper, “Association between FOXP3 and OX40 Expression in Adult T-cell Leukemia Cells,” appropriately reflects the content.
To establish a statistically significant correlation, further experiments with a larger number of ATL patient samples are certainly required. These experiments are currently in preparation, and the results will be reported in a future paper. At this stage, we wished to present the new finding that there may be a consistent relationship between FOXP3 and OX40 expression based on PBMCs from three independent ATL patients (Figure 1).
Reviewer 4 Report
Comments and Suggestions for Authors
Dear Authors,
I have reviewed the revised manuscript entitled “Correlation between FOXP3 and OX40 Expression in Adult T-cell Leukemia Cells” submitted by Mizuguchi et al. I am pleased to note that the authors have largely addressed the comments I provided and authors have made significant improvements in the revised manuscript.
The revisions have clarified the methodology, strengthened the discussion, and enhanced the overall quality and readability of the work. Though the research design has some limitations due to the nature of the samples, based on these improvements, I recommend that the manuscript be accepted for publication.
Sincerely,
Author Response
Comment #4-1
Dear Authors,
I have reviewed the revised manuscript entitled “Correlation between FOXP3 and OX40 Expression in Adult T-cell Leukemia Cells” submitted by Mizuguchi et al. I am pleased to note that the authors have largely addressed the comments I provided and authors have made significant improvements in the revised manuscript.
The revisions have clarified the methodology, strengthened the discussion, and enhanced the overall quality and readability of the work. Though the research design has some limitations due to the nature of the samples, based on these improvements, I recommend that the manuscript be accepted for publication.
Sincerely,
Response #4-1
We sincerely appreciate the reviewer’s constructive comments.